# DeeperForward: Enhanced Forward-Forward Training for Deeper and Better Performance

**Liang Sun**[1,†], **Yang Zhang**[1,†,*], **Weizhao He**[1], **Jiajun Wen**[1], **Linlin Shen**[1,2,3], **Weicheng Xie**[1,2,3]

[1]Computer Vision Institute, School of Computer Science & Software Engineering, Shenzhen University
[2]National Engineering Laboratory for Big Data System Computing Technology, Shenzhen University
[3]Guangdong Provincial Key Laboratory of Intelligent Information Processing
{sunliang, heweizhao}2022@email.szu.edu.cn,
{yangzhang, wenjiajun, llshen, wcxie}@szu.edu.cn
**Code: https://github.com/tobysunsun/deeperforward**

## Abstract

While backpropagation effectively trains models, it presents challenges related to bio-plausibility, resulting in high memory demands and limited parallelism. Recently, Hinton (2022) proposed the Forward-Forward (FF) algorithm for high-parallel local updates. FF leverages squared sums as the local update target, termed goodness, and decouples goodness by normalizing the vector length to extract new features. However, this design encounters issues with feature scaling and deactivated neurons, limiting its application mainly to shallow networks. This paper proposes a novel goodness design utilizing **layer normalization** and **mean goodness** to overcome these challenges, demonstrating performance improvements even in 17-layer CNNs. Experiments on CIFAR-10, MNIST, and Fashion-MNIST show significant advantages over existing FF-based algorithms, highlighting the potential of FF in deep models. Furthermore, the model parallel strategy is proposed to achieve highly efficient training based on the property of local updates.

## 1 Introduction

Backpropagation (BP) (Rumelhart et al., 1986) has achieved significant success, serving as the prevailing paradigm for training complex structures like ResNet (He et al., 2016) and Transformers (Vaswani et al., 2017). However, no compelling evidence supports such a mechanism existing in the brain, challenging the biological plausibility of BP. Critical challenges within BP consist of weight transport (Grossberg, 1987), non-local (Whittington & Bogacz, 2019), freezing activity, and update locking problems (Jaderberg et al., 2017; Czarnecki et al., 2017). The *weight transport problem* arises from reusing the same path in forward and backward passes. The *non-local problem* arises from global objective loss, while the brain relies on local signals for updates. The *freezing activity problem* and *update locking problem* contradict the real-time property in neural systems. Freezing activity involves maintaining intermediate states, leading to increased memory demands. The update locking problem prevents any update until all layers are activated, reducing parallelism in practice.

To tackle these challenges, various brain-inspired training methods (Ororbia, 2023) have been developed to formulate a comprehensive theory of inference and learning in a biologically plausible manner (Lillicrap et al., 2016; Nøkland, 2016; Dellaferrera & Kreiman, 2022; Ororbia et al., 2023; Hinton, 2022). Several of these methods are depicted in Figure 1. A recent breakthrough is the Forward-Forward (FF) algorithm (Hinton, 2022), as depicted in Figure 1(d). FF employs the squared sum of outputs, termed *goodness*, and fixes the output vector length via dividing by its vector length, thereby decoupling goodness within the output features and compelling subsequent layers to learn new features. Mathematically, the output's length and direction correspond to goodness and features. However, this design has limitations that confine current layer-wise FF studies to shallow models. The primary reasons why FF fails to achieve performance improvements in deeper networks are as follows:

---

[†]Equal Contribution: Liang Sun and Yang Zhang.
[*]Corresponding author: Yang Zhang.

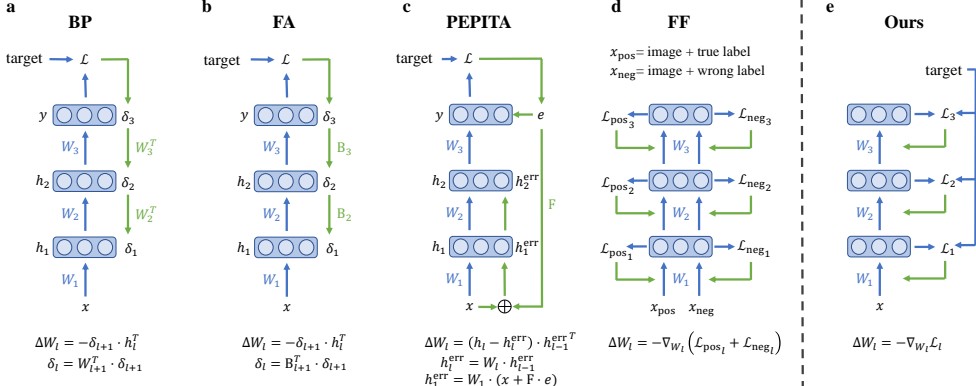

Figure 1: Comparison of several training methods. (a) BP employs traditional forward and backward passes represented by blue and green arrows respectively. (b) Feedback alignment (FA) uses an alternative backward pass for error passes. (c) PEPITA uses two forward passes based on input perturbation. (d) FF is implemented with two forward passes on positive data and negative data, respectively. (e) Ours simplifies the learning process by using a single forward pass.

**Feature scaling** Normalization by vector length is uncommon in image classification tasks as it does not ensure that features exhibit similar characteristics, such as identical means and standard deviations. To address this, layer normalization (Ba et al., 2016) can be applied to the input vector, but this leads to redundant normalization and compromises the mathematical significance of output direction as a feature. Consequently, CwComp (Papachristodoulou et al., 2024) employs batch normalization as a substitute. However, this approach fails to decouple goodness and leaks goodness to the next layer, hindering deeper layers from learning new features and causing overfitting.

**Deactivated neurons** Square goodness is highly sensitive to outliers, which can dominate and deactivate most neurons, distorting feature representation. Moreover, deactivated neurons do not contribute to weight updates during gradient calculations, leading to features represented by a limited subset of neurons and causing feature loss in deeper layers.

In this paper, DeeperForward is proposed to address the mentioned issues by redesigning goodness and features to better suit deep networks. We also enhance the convolution structure proposed by CwComp (Papachristodoulou et al., 2024), enabling effective training of FF in deeper CNNs. Our main contributions are as follows:

- We adopt the more widely used layer normalization(Ba et al., 2016) to ensure a fixed mean and standard deviation of the output, replacing normalization based on vector length and effectively addressing feature scaling and redundant normalization issues.
- Exploiting the property of layer normalization that maintains a mean of zero, we propose using mean goodness as an alternative to squared goodness, thereby facilitating the decoupling of goodness for enhanced feature extraction. This approach also ensures that weight updates are not hindered by deactivated neurons, allowing for the learning of richer features.
- Based on the characteristics of layer-wise local updates, we introduce a model parallel strategy that significantly enhances training efficiency on multiple GPUs.
- Our method enhances FF to achieve improved performance in deeper networks. Experimental results indicate that our approach, utilizing a 17-layer CNN, outperforms existing layer-wise FF-based methods on CIFAR-10, MNIST, and Fashion-MNIST, achieving substantial performance gains, particularly an 8.11% improvement on CIFAR-10

## 2 RELATED WORK

### 2.1 CONVENTIONAL BRAIN-INSPIRED LEARNING RULES

Hebbian learning (Hebb, 2005; Gerstner et al., 2014) updates synaptic plasticity determined by pre- and post-synaptic neuron states (Löwel & Singer, 1992). Based on the Hebbian rule, a neural coding

framework was proposed for learning generative models using the predictive coding (Ororbia & Kifer, 2022; Rao & Ballard, 1999). SoftHebb (Journé et al., 2023) proposes an algorithm based on theory for Hebbian learning in soft winner-take-all (WTA) networks. Hebbian learning is considered a basic bio-plausible method with no target.

In target-based methods, feedback alignment (Lillicrap et al., 2016) and direct feedback alignment (Nøkland, 2016) replace backpropagation weights with a fixed random matrix to establish alternative error feedback connections, as shown in 1(b). Weight mirror (Akrout et al., 2019) adjusts the feedback connection matrix, equivalent to the transport weight matrix. However, these methods still rely on global error. Target propagation (TP) (Bengio, 2014; Bartunov et al., 2018) and difference target propagation (DTP) (Lee et al., 2015; Ernoult et al., 2022) set local targets as the goal for local updates. Local representation alignment (LRA) (Ororbia et al., 2023) addresses the asymmetry problem through top-down signal transmission with Hebbian-like rules, further solving the non-local problem. To update unlocking, decoupled greedy learning (Belilovsky et al., 2020) optimizes a joint training objective to decouple the layer training with auxiliary networks. Avoiding using backward passes, a forward propagation training method through time is proposed for recurrent neural networks (Kag & Saligrama, 2021). PEPITA (Dellaferrera & Kreiman, 2022) achieves local updates by perturbing inputs with the error and employs a Hebbian-like rule based on two forward passes with a fixed feedback matrix, as shown in Figure 1(c). Despite these advancements, they partially suffer from the update locking problem.

## 2.2 BACKGROUND OF FORWARD-FORWARD ALGORITHM

Inspired by Boltzmann machines (Hinton et al., 1986) and noise contrastive estimation (NCE) (Gutmann & Hyvärinen, 2010), the Forward-Forward algorithm (FF) (Hinton, 2022) introduces a greedy learning scheme via two forward passes, as shown in Figure 1(d), tackling the mentioned bio-implausible problems. FF uses the length of the output vector as a measure of *goodness*, where goodness represents the score of positive data. Decoupling goodness from the output features is important to prevent subsequent layers from relying solely on previous goodness. Therefore, FF extracts features by normalizing the vector length, denoted as,

$$\boldsymbol{y} = \mathrm{ReLU}(\boldsymbol{Wx}), \tag{1}$$

$$g = \sum_i y_i^2, \quad \boldsymbol{z} = \frac{\boldsymbol{y}}{\sqrt{\frac{1}{N}g + \epsilon}}, \tag{2}$$

$$\Delta W_{ij} = 2 x_i y_j \frac{\partial \mathcal{L}}{\partial g}, \tag{3}$$

where $\boldsymbol{x}$ denotes the input, $\boldsymbol{y}$ represents the output after ReLU (Glorot et al., 2011) with $N$ elements, $\boldsymbol{W}$ is the weight matrix, $y_i$ denotes the element of the vector $\boldsymbol{y}$ of a hidden layer, and $g$ denotes goodness. The features $\boldsymbol{z}$ is the unit vector of $\boldsymbol{y}$. $\epsilon$ is a small constant. $\Delta W_{ij}$ denotes the weight update term and $\mathcal{L}$ is the loss function. The image with a real label is regarded as positive data for optimizing to reach a high goodness value in each layer, and vice versa. During inference, an image entails computing the goodness of each label and selecting the highest one through several iterations. The preliminary study of FF only works on small networks without weight-sharing structures.

Recently, several works have proposed some advanced FF-related algorithms. Symmetric backpropagation-free contrastive learning with FF (SymBa) (Lee & Song, 2023) enhances performance through a gradient-symmetric contrastive loss and a novel label embedding scheme. The predictive Forward-Forward algorithm (PFF) (Ororbia & Mali, 2023) integrates FF with predictive coding presenting a promising brain-inspired algorithm for classifying, reconstructing, and synthesizing data patterns. However, these approaches are still limited to models without weight-sharing structures. The cascaded forward (CaFo) algorithm (Zhao et al., 2023) utilizes a series of random fixed convolutional kernels as the backbone and cascades a fully connected classifier for each kernel. However, it merely updates the classifiers, leaving the kernels unchanged. Forward-Forward contrastive learning (FFCL) (Ahamed et al., 2023) introduces contrastive learning for convolutional models based on FF. However, this approach still prefers extra training by global errors. Recently, convolutional channel-wise competitive learning (CwComp) (Papachristodoulou et al., 2023; 2024) successfully extends FF into CNNs by grouping the features by channels for each class, and using a loss function inducing competitive learning between class-specific features. Despite the advancements, these methods focus on shallow networks within 4 layers. Currently, Trifecta (Dooms et al.,

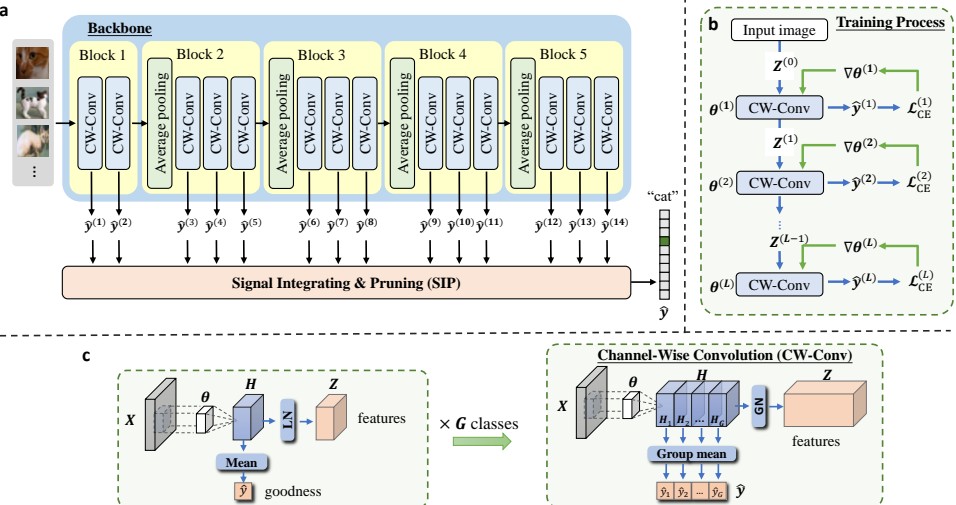

Figure 2: Overview of DeeperForward. (a) Example network architecture for DeeperForward, including *backbone* and *Signal Integrating and Pruning* module. A VGG-like architecture is displayed as an instance. (b) The training scheme of DeeperForward. (c) Modified channel-wise convolution (CW-Conv) from CwComp based on mean goodness.

2023) employs a two-layer block-wise backpropagation approach to replace single-layer updates in a 12-layer CNN, using batch normalization. However, this integration with backpropagation still presents bio-plausibility issues, diminishing parallelism and contradicting the motivations behind the FF algorithm. Both Trifecta and CwComp facilitate easier training by leaking goodness, which can result in potential overfitting in deeper networks.

## 3 METHODOLOGY

This paper introduces DeeperForward, which extends the FF algorithm to 17-layer CNNs through a novel goodness design. This approach addresses the bio-plausibility issues of backpropagation and overcomes the limitations of FF concerning model size. Figure 2 illustrates the overall framework of our method, including the architecture and training approach. The details of the new goodness design are presented in Section 3.1. The network architecture is discussed in Section 3.2, while the training process and advanced strategies for DeeperForward are outlined in Section 3.3.

### 3.1 MEAN GOODNESS

FF uses squared goodness and normalization of the length, as described in Eq. 2. This method suffers from issues related to feature scaling, deactivated neurons, and redundant normalization, resulting in suboptimal performance in deep networks. Although CwComp (Papachristodoulou et al., 2024) improves performance using squared goodness and batch normalization, it leaks goodness information, leading to overfitting in deeper networks. Considering these factors, we adopt widely used layer normalization for better feature scaling with identical mean and standard deviations. To decouple goodness through normalization, we utilize the mean as goodness, leveraging the property of layer normalization that produces an output with a mean of zero. Furthermore, mean goodness ensures that deactivated neurons do not hinder updates. The specific formula is as follows:

$$\boldsymbol{y} = \mathrm{ReLU}(\boldsymbol{W}\boldsymbol{x}), \tag{4}$$

$$g = \sum_i y_i, \quad \boldsymbol{z} = \frac{\boldsymbol{y} - g}{\sqrt{\sigma^2 + \epsilon}}, \tag{5}$$

$$\Delta W_{ij} = C x_i \frac{\partial \mathcal{L}}{\partial g}, \tag{6}$$

where $x$ denotes the input, $y$ represents the output after ReLU (Glorot et al., 2011), $W$ is the weight matrix, $g$ indicates goodness, $\sigma$ is the standard deviation, $z$ refers to the output features, $\Delta W_{ij}$ is the weight update term, $\mathcal{L}$ is the local loss function, and $C$ is a constant.

From Eq. 5, it is evident that the output distribution $z$ maintains a mean of zero, effectively eliminating goodness. This also ensures that the features share a similar distribution, addressing the feature scaling issue. During weight updates, mean goodness (Eq. 6) allows for updates even when the output neuron $y_j$ is zero, unlike squared goodness (Eq. 3), thereby solving the deactivated neurons problem.

## 3.2 ARCHITECTURE FOR DEEPERFORWARD

The architecture for DeeperForward, as illustrated in Figure 2(a), incorporates a modified classical CNN backbone, exemplified by the VGG-like model (Simonyan & Zisserman, 2014). It incorporates a convolutional structure that combines channel-wise convolution (CW-Conv) with mean goodness, along with a *Signal Integrating and Pruning* (SIP) module to obtain the final results.

**Channel-Wise Convolution with Mean Goodness**  To incorporate mean goodness into CNNs, combining convolution with mean goodness involves simply obtaining the output mean as goodness, followed by layer normalization to facilitate feature extraction. Formally, the goodness $\hat{y}$ and representation $Z$ are defined as:

$$\hat{y} = \frac{1}{HWC} \sum_{h \in H} h, \quad Z = \text{LayerNorm}(H), \tag{7}$$

where $H \in \mathbb{R}^{H \times W \times C}$ denotes the hidden states after the convolution with ReLU (Glorot et al., 2011). $\hat{y} \in \mathbb{R}$ indicates the goodness, that is, the mean of $H$. The representation output $Z$ is $H$ going through layer normalization.

In multi-class tasks, we optimize the channel-wise convolution (CW-Conv) structure from CwComp (Papachristodoulou et al., 2023; 2024), combining it with our mean goodness to obtain goodness scores for all classes through a single inference. The outputs are evenly grouped by channel, with each group representing a class. Goodness is calculated for each group, followed by individual layer normalization to extract features, effectively implementing group normalization (Wu & He, 2018) on the entire output, as illustrated in Figure 2(c). Formally, the channel-wise convolution with mean goodness for $G$ classes can be described as:

$$\hat{y}_i = \frac{G}{HWC} \sum_{h \in H_i} h, \quad i = 1, 2, ..., G;$$

$$\hat{y} = [\hat{y}_1, \hat{y}_2, ..., \hat{y}_G], \tag{8}$$

$$Z = \text{GroupNorm}(H; G), \tag{9}$$

where $H_i \in \mathbb{R}^{H \times W \times \frac{C}{G}}$ and $\hat{y}_i$ denotes the hidden states and goodness for the $i$-th class, and $\hat{y}$ stands for classification scores. $\text{GroupNorm}(H; G)$ represents the group normalization of hidden states $H$ by $G$ groups. $Z$ stores the representation feature maps.

Compared to our method, CwComp performs classification training directly on the outputs after batch normalization without decoupling goodness, resulting in goodness leakage to the next layer and leading to overfitting in deeper layers.

**Backbone**  The backbone is derived from the classical CNNs, leveraging their well-established structural advantages. We substitute the general convolutional kernels with CW-Conv modules to generate classification results locally. The representation from each CW-Conv module serves as the input for the next layer, denoted as $Z^{(l)}$ where $l$ signifies the layer number. The local classification result at the $l$-th layer is represented as $\hat{y}^{(l)}$. In particular, the channel size of kernels must be a multiple of the class count. Furthermore, to maintain approximate zero mean of the representation, we adopt average pooling for downsampling, instead of max pooling, as the latter tends to increase the mean value. In this architecture, each layer produces a classification score using CW-Conv. Moreover, the experiments in Appendix E reveal that the CW-Conv outperforms the fully connected (FC) layer in terms of performance. Consequently, the final FC layer is needless.

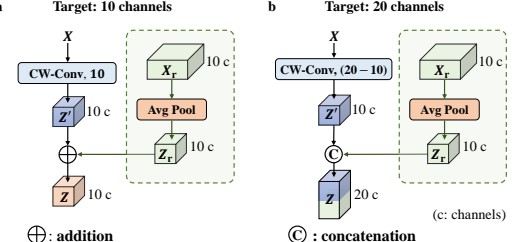

Figure 3: Residual structures: (a) Addition type for the shortcut channels match the target channels. (b) Concatenation type for shortcut channels differing from the target channels.

**Residual Structure** Residual structures are traditionally employed to facilitate error backpropagation by providing shortcuts for easier learning. In FF, it enables the integration of features at different levels, enriching the representational capacity of deep networks. To adapt to the FF, we implement two parameter-free residual structures, the *addition* and *concatenation* types, as alternatives to the original parameterized versions, as illustrated in Figure 3. To match the spatial dimensions, we employ average pooling to the shortcut, as shown below, for downsampling.

$$\mathbf{Z}_{\mathrm{r}} = \begin{cases} \mathrm{AvgPool}(\boldsymbol{X}_{\mathrm{r}}), & (H_{\mathrm{r}}, W_{\mathrm{r}}) \neq (H, W), \\ \boldsymbol{X}_{\mathrm{r}}, & (H_{\mathrm{r}}, W_{\mathrm{r}}) = (H, W), \end{cases} \tag{10}$$

where $\mathbf{Z}_{\mathrm{r}} \in \mathbb{R}^{H \times W \times C_{\mathrm{r}}}$ denotes the shortcut feature map after spatial dimension adjustment, and $\mathrm{AvgPool}(\cdot)$ is the average pooling operation to adjust $\boldsymbol{X}_{\mathrm{r}}$ from $(H_{\mathrm{r}}, W_{\mathrm{r}})$ to $(H, W)$.

Two types of residual structures are adapted in different scenarios. If the shortcut matches the channel of target feature maps, the addition type is employed. Otherwise, the concatenation type is used. As shown in Figure 3, two residual structures can be summarized as:

$$\mathbf{Z} = \begin{cases} F(\boldsymbol{X}; C) + \mathbf{Z}_{\mathrm{r}}, & C = C_{\mathrm{r}}, \\ \mathrm{Concat}(F(\boldsymbol{X}; C - C_{\mathrm{r}}), \mathbf{Z}_{\mathrm{r}}), & C \neq C_{\mathrm{r}}, \end{cases} \tag{11}$$

where $F(\boldsymbol{X}; C)$ stands for the CW-Conv with $C$ channels output from input $\boldsymbol{X}$, and $\mathbf{Z}_{\mathrm{r}}$ is the feature maps from shortcut. $\mathrm{Concat}(\cdot, \cdot)$ denotes the concatenation operation on channel dimension. $\mathbf{Z}$ represents the final representation output. Particularly, the number of convolution channels is reduced to $C - C_{\mathrm{r}}$ in concatenation type to ensure the channel of final output satisfies the target.

**Signal Integrating and Pruning Module** Inspired by synaptic pruning (Chechik et al., 1998; Neniskyte & Gross, 2017), where the brain forms excess synapses and then eliminates redundancies, we propose the *Signal Integrating and Pruning* (SIP) module. The FF accumulates local goodness to obtain the final result, with experiments showing that the last three layers perform best on the test set. Similarly, we separate a subset of data from the training set, leave it untrained, and evaluate accuracy on this subset to select the best layer combination, avoiding direct testing on the test set. However, for a deep model with $L$ layers, there are $2^L$ combinations. To reduce complexity, we simplify the rule to accumulating layers between a chosen start layer and an end layer, reducing the combinations to $L(L+1)/2$. The SIP module with $L$ layers can be described as:

$$\hat{\boldsymbol{y}} = \sum_{l=S}^{E} \hat{\boldsymbol{y}}^{(l)}, \quad 0 < S \leq E \leq L, \tag{12}$$

where $\hat{\boldsymbol{y}}^{(l)}$ denotes classification scores from the $l$-th layer, and $\hat{\boldsymbol{y}}$ is the final result. $S, E \in \mathbb{Z}$ are integers and range from 1 to $L$, representing the start and end layers to be accumulated. After selection, layers beyond the end layer are no longer used and can be pruned.

### 3.3 DEEPERFORWARD TRAINING SCHEME

**Training Scheme** We present DeeperForward, a training strategy that optimizes the classification result at each layer through a single forward pass, relying solely on the local input-output states.

Figure2(b) depicts the training procedure of DeeperForward. Local optimization leverages the classification results from CW-Conv as the local target. It utilizes a local cross-entropy loss for each layer to generate the update signal, preventing error transportation across layers. The local optimization can be formulated as:

$$\mathcal{L}_{\mathrm{CE}}^{(l)}(\hat{\boldsymbol{y}}^{(l)}, \boldsymbol{y}) = -\sum_{i=1}^{G} y_i \log(\mathrm{softmax}(\hat{y}_i^{(l)})),$$
$$\nabla \boldsymbol{\theta}^{(l)} = \nabla_{\boldsymbol{\theta}^{(l)}} \mathcal{L}_{\mathrm{CE}}^{(l)}(\hat{\boldsymbol{y}}^{(l)}, \boldsymbol{y}), \qquad (13)$$

where $\hat{\boldsymbol{y}}^{(l)}$ and $\boldsymbol{y}$ denote the local classification result and the real label with $G$ classes. $\mathcal{L}_{\mathrm{CE}}^{(l)}$ denotes the cross-entropy loss, while $\nabla \boldsymbol{\theta}^{(l)}$ is the update of weights at the $l$-th layer, and $\nabla_{\boldsymbol{\theta}^{(l)}} \mathcal{L}_{\mathrm{CE}}^{(l)}(\hat{\boldsymbol{y}}^{(l)}, \boldsymbol{y})$ is the gradient of $\mathcal{L}_{\mathrm{CE}}^{(l)}$ with respect to the kernel weights $\boldsymbol{\theta}^{(l)}$. The local optimization process solely relies on the input and classification result of the individual CW-Conv. DeeperForward is compatible with general gradient-based optimizers, such as Adam (Kingma & Ba, 2014). Owing to the local learning process, there is no need to store intermediate states, eliminating the freezing activity problem and the update locking problem. Additionally, the non-local problem and weight transport problem are addressed by the local loss optimization and the forward training scheme.

**Model Parallel Strategy** Our method enables a model parallel strategy based on the parallelism of FF mentioned in (Aktemur et al., 2024), as illustrated in Figure 4. Our strategy treats each convolutional layer as an independent component in the pipeline. Once a convolutional layer processes a batch of data, it passes the results to the next group, allowing the next batch to be processed without waiting for the entire network to complete. As shown in Figure 4 (a), this approach enables simultaneous processing of multiple batches across different layers, achieving high parallelism.

Figure 4(b) illustrates an implementation example using multithreading techniques in a multi-GPU setup. Our strategy assigns an independent thread for each convolutional layer to update, utilizing first-in-first-out(FIFO) queues for data transfer. Different threads can be allocated to various GPUs, enabling model parallelism. Compared to the commonly used distributed data-parallel (DDP) (Li et al., 2020) technique in backpropagation, our approach offers several advantages for improved efficiency: (i) Each GPU does not need to store the entire network, and (ii) Data transfer between GPUs occurs only between layers on different GPUs, rather than across the entire network. Details of implementation are in Appendix G.

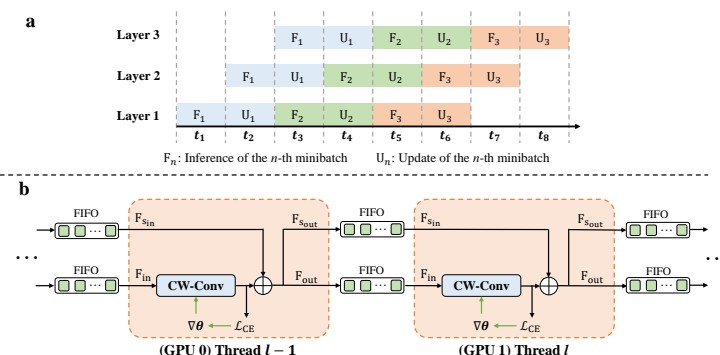

Figure 4: Model parallel strategy. (a) A pipeline program (the same color indicates operations on the same minibatch of data). (b) An implementation example based on the multi-threading technique.

**Memory-saving Strategy** Due to update locking problems, BP requires a large amount of memory to store intermediate states throughout the process. Without limitation of update locking, this strategy achieves memory savings by promptly releasing memory after each layer's computation. The memory saving strategy is a layer-by-layer update strategy, which consists of following steps: (1) Perform computation and update weights in current layer. (2) Pass the output to the next layer. (3) Release all intermediate states from the memory used by the current layer. (4) Repeat steps (1)-(3) layer by layer. More details are shown in Appendix H. This memory-saving strategy is particularly suitable for scenarios with constrained memory resources, such as edge computing.

Table 1: Classification on CIFAR10, MNIST, and F-MNIST, evaluating performance compared to BP and FF-related algorithms. Measurements of mean and standard deviation are for five trial runs. *: Reproduced results. †: With data augmentation. ‡: With block-wise backprop.

| Type | Method | Arch. | #Layer | CIFAR10 | MNIST | F-MNIST |
|---|---|---|---|---|---|---|
| non-FF | PEPITA | CNN | 2 | $52.57 \pm 0.36$ | $98.01 \pm 0.09$ | - |
| | DTP | CNN | 6 | $89.38 \pm 0.20$ | $98.93 \pm 0.04$ | $\mathbf{90.35 \pm 0.11}$ |
| | recLRA | CNN | 18 | $\mathbf{93.58}$ | 98.18 | 88.13 |
| | SoftHebb | SoftHebb | 4 | $80.31 \pm 0.14$ | $\mathbf{99.35 \pm 0.03}$ | - |
| | $F^3$ | MLP | 2 | $46.04 \pm 0.18$ | $97.16 \pm 0.10$ | - |
| | SP | CNN | 8 | 92.4 | - | - |
| Block-wise BP | HPFF | CNN | 110 | 91.04 | - | - |
| | SEDONA | CNN | 152 | 93.87 | - | - |
| | BWBPF | CNN | 152 | $\mathbf{95.52}$ | - | - |
| BP | ResNet18-BP* | CNN | 18 | $94.03 \pm 0.11$† | $99.58 \pm 0.02$ | $93.78 \pm 0.06$ |
| FF | FF | MLP | 4 | 59.00 | 98.69 | - |
| | SymBa | MLP | 3 | 59.09 | 98.58 | - |
| | CaFo | CNN | 3 | 67.43 | 98.80 | - |
| | CwComp | CNN | 4 | $78.11 \pm 0.44$ | $99.42 \pm 0.08$ | $\mathbf{92.31 \pm 0.32}$ |
| | **TinyCNN-ours** | CNN | 4 | $\mathbf{79.49 \pm 0.29}$ | $\mathbf{99.50 \pm 0.05}$ | $91.83 \pm 0.06$ |
| FF | Trifecta‡ | CNN | 12 | $83.51 \pm 0.78$ | $99.58 \pm 0.06$ | $91.44 \pm 0.49$ |
| | CwComp* | CNN | 14 | $75.28 \pm 0.54$ | $99.27 \pm 0.09$ | $91.79 \pm 0.47$ |
| | **CNN-ours** | CNN | 14 | $81.76 \pm 0.30$ | $\mathbf{99.65 \pm 0.02}$ | $92.44 \pm 0.08$ |
| | **ResNet-ours** | CNN | 17 | $\mathbf{86.22 \pm 0.17}$ | $99.63 \pm 0.04$ | $\mathbf{93.13 \pm 0.13}$ |

Table 2: Classification on CIFAR100.

| | ResNet-BP | ResNet-ours | ResNet-CHx3-ours |
|---|---|---|---|
| **Accuracy** | $58.01 \pm 0.48$ | $53.09 \pm 0.79$ | $60.28 \pm 1.02$ |

# 4 EXPERIMENT

## 4.1 DATASETS AND EXPERIMENT SETTINGS

To fully validate the effectiveness of DeeperForward, we conduct experiments on 3 datasets: MNIST (LeCun et al., 1998), Fashion-MNIST (F-MNIST) (Xiao et al., 2017), and CIFAR10 (Krizhevsky et al., 2009) without any data augmentation. Specifically, the training sets of MNIST and F-MINST are separated into two groups, 50,000 and 10,000 samples. The former group is used for training and the latter group is used for pruning by *Signal Integrating and Pruning* (SIP) module. Similarly, CIFAR10's training set is split into 45,000 and 5,000 samples. All the samples in the datasets are resized to $32 \times 32$ pixels. Hyperparameters setting is detailed in Appendix B. Our experiments are executed on 4 Nvidia GTX Titan X GPUs (12GB).

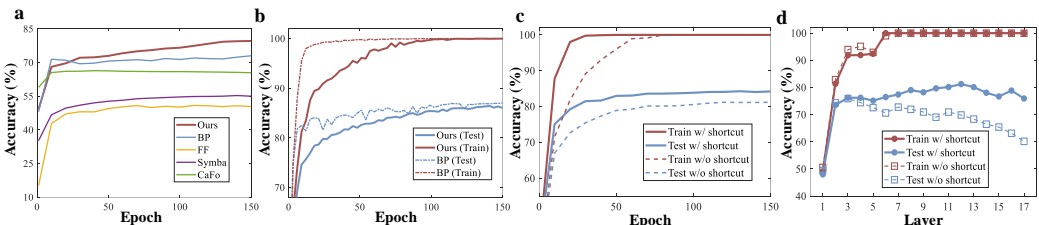

Figure 5: Performances on CIFAR10. (a) Shallow networks, compared to BP and FF-based methods. (b) Deep networks, compared to BP without data augmentation. (c,d) Comparison experiments for residual structures on CIFAR10 without dropout. (c) Model performance during training. (d) The accuracy of each layer after training for 150 epochs.

## 4.2 COMPARISONS OF DIFFERENT METHODS

We employ three CNN models to evaluate our method: a 4-layer tiny CNN, a VGG-like 14-layer CNN, and a 17-layer ResNet-like CNN (He et al., 2016), as detailed in the Appendix A. Our comparisons encompass both non-FF and FF methods for a comprehensive analysis. Non-FF brain-inspired methods include advanced BP-free approaches such as PEPITA (Dellaferrera & Kreiman, 2022), DTP (Ernoult et al., 2022), rec-LRA (Ororbia et al., 2023), SoftHebb (Journé et al., 2023), $F^3$ (Flügel et al., 2023), and Signal Propagation(SP) (Kohan et al., 2024). We also compare with block-wise BP method: HPFF(Su et al., 2024), SEDONA(Pyeon et al., 2021), and BWBPF(Cheng et al., 2024). In the FF methods, we consider FF (Hinton, 2022), SymBa (Lee & Song, 2023), CaFo (Zhao et al., 2023), Trifecta (Dooms et al., 2023), and CwComp (Papachristodoulou et al., 2024). Since layer-wise FF-based methods operate on shallow networks, we reproduce and extend CwComp (Papachristodoulou et al., 2024) into the same 14-layer CNNs for comparison. Results are summarized in Table 1. Our method outperforms FF-based methods in CIFAR10 and MNIST with shallow networks. The training curves on CIFAR10 with FF-based methods in shallow networks are shown in Figure 5(a). As we extend to 14 and 17 layers, our performance improves, whereas CwComp (Papachristodoulou et al., 2024) exhibits overfitting, leading to performance decline. Therefore, our method extended the capability of FF to train in deeper models. The results indicate that our design of mean goodness enhances the performance of FF, making it more suitable for deep CNN models. However, FF-based methods train greedily through classification objectives at each layer, indirectly extracting features. This results in weaker feature learning capabilities compared to BP, which directly learns intermediate layer features. Although Figure 5(b) shows that our method's performance is close to BP without data augmentation, the improvement is limited after data augmentation, as detailed in Appendix C, leading to a larger gap with BP. These points are also directions worth exploring further.

Furthermore, we conduct a more challenging experiment on CIFAR100, as shown in Table 2. ResNet-CHx3 is a variant of ResNet with triple the number of channels. Table 2 highlights the disparity between DeeperForward and BP on ResNet. The significant improvement of ResNet-CHx3 indicates that an inadequate allocation of neurons to each class results in a sharp decline in performance.

## 4.3 PERFORMANCE OF SIGNAL INTEGRATING AND PRUNING MODULE

Table 3: Performance on Signal Integrating and Pruning (SIP) using a 17-layer ResNet. $(Start, End)$ denotes the selected layers by SIP, where $Start$ and $End$ represent start layer and end layer.

|  | CIFAR10 | MNIST | F-MNIST |
|---|---|---|---|
| ALL LAYERS | 86.45 | 99.68 | 93.08 |
| AFTER SIP | 86.51 | 99.67 | 93.23 |
| $(Start, End)$ | (2,17) | (2,11) | (3,16) |

We validate our pruning strategy using the Signal Integrating and Pruning (SIP) module by comparing the performance with a similar strategy in FF (Hinton, 2022) that accumulates all the goodness as the final result. In Table 3, we select the best trial results using ResNet for the SIP experiment comparison, showing that SIP can improve performance in most cases. Interestingly, experiments on simpler tasks such as MNIST tend to retain fewer layers compared to more challenging tasks like CIFAR10. This observation shows the ability to adapt its depth based on the complexity of the task.

## 4.4 ABLATION STUDY

Table 4: Ablation study on CIFAR-10, showing the mean performance from five experimental trials.

| MEAN | SIP | RESIDUAL | ACCURACY |
|---|---|---|---|
|  | ✓ | ✓ | 79.38 |
| ✓ |  |  | 81.02 |
| ✓ | ✓ |  | 81.16 |
| ✓ |  | ✓ | 86.08 |
| ✓ | ✓ | ✓ | 86.22 |

This method introduces mean goodness, the Signal Integrating and Pruning (SIP) module, and a non-learned residual structure to optimize performance. To evaluate the contributions of each component, we conducted ablation experiments using a 17-layer ResNet architecture on CIFAR10, averaging results from five trials, as shown in Table 4. When mean goodness is omitted, we utilize

squared goodness and normalization of the vector length. For comparison with SIP, we directly sum all layers as the final output. The removal of the residual structure involves excluding the shortcut connections. To provide a more comprehensive analysis, Appendix I discusses the differences in deactivated neurons between mean and square goodness.

The experimental results show that mean goodness achieves a substantial performance increase of 6.84% compared to squared goodness within the same network. The SIP module provides a slight performance boost while allowing for optimization of network size. The residual structure significantly enhances performance by integrating features at various levels, resulting in a more comprehensive feature representation. Figures 5 (c) and (d) analyze the training curves and local classification performance with and without the residual structure, demonstrating that the residual connections facilitate improved learning in deeper layers.

### 4.5 Parallel Performance of DeeperForward

We evaluate the performance of model parallel strategy through time-consumption training on CIFAR10, using 1, 2, and 4 GPUs. In multi-GPU case, layers are evenly grouped and assigned to different GPUs. As a point of comparison, we use BP with the widely adopted distributed data parallel (DDP) (Li et al., 2020) as a baseline. As shown in Table 5, our method outperforms BP with DDP in terms of training time. Notably, our approach achieves a higher speedup with 2 GPUs, as inter-GPU communication occurs only between layers on different devices, unlike DDP, where communication involves the entire network. However, with 4 GPUs, the speedup is lower than DDP. We observed a drop in GPU utilization, caused by imbalanced computation across layers, leading to pipeline program bottlenecks. Future work could explore advanced pipeline techniques for optimization. This study demonstrates the feasibility and potential of model parallelism in our method.

Table 5: Training time per epoch on CIFAR10 (Speedup rate relative to 1 GPU in parentheses).

| Method | 1 GPU | 2 GPUs | 4 GPUs |
|---|---|---|---|
| BP-DDP | 51.98s ($1.0\times$) | 32.70s ($1.59\times$) | 19.92s (**2.61**$\times$) |
| Ours | **36.38s** ($1.0\times$) | **20.77s** (**1.75**$\times$) | **14.68s** ($2.48\times$) |

Moreover, training on CIFAR10 with a batch size of 128 using the memory-saving strategy consumes a minimum of 618.64MB of memory in practice, while BP in ResNet18 requires 1314.49MB.

Additionally, we experiment with deeper ResNet models with 33 and 100 layers but do not observe significant performance improvements, as detailed in Appendix D. Moreover, Appendix E provides a comparison of classification performance between different convolutional layers and fully connected layers. Appendix F presents t-SNE (Van der Maaten & Hinton, 2008) visualizations of the results on the MNIST dataset.

## 5 Conclusion

This paper presents the DeeperForward algorithm, extending the Forward-Forward approach to deeper networks with significant performance enhancements. We introduce a novel goodness design, combining mean goodness and layer normalization, which addresses key issues in the effective training of deep networks: feature scaling, redundant normalization, and deactivated neurons. Additionally, we propose a model parallel strategy to significantly improve training efficiency and a memory-saving strategy suitable for resource-constrained environments. Experimental results demonstrate that our method substantially enhances the depth and performance of FF-based algorithms, highlighting the potential of FF in terms of performance and parallelism.

**Limitations.** DeeperForward, similar to FF, relies solely on classification information for learning, lacking direct representation learning capabilities. This results in slower convergence and weaker generalization. Future research should focus on enhancing feature extraction capabilities to address these limitations. Additionally, as the number of categories increases, the convolutional structure grows, making it challenging to implement on extensive datasets. Future work should aim to develop more general structures that avoid excessively large models and multiple forward passes in FF.

ACKNOWLEDGEMENTS

This work is supported by the National Natural Science Foundation of China under Grant 62176163; the Shenzhen Higher Education Stable Support Program General Project under Grant 20231120175215001; and the Science and Technology Foundation of Shenzhen under Grant JCYJ20210324094602007.

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

## A  NETWORK ARCHITECTURES

We perform experiments using the network architectures depicted in Figure 6, which include Tiny-CNN, CNN, and ResNet. TinyCNN is a shallow 4-layer CNN designed for performance comparison with FF-related methods that are effective in shallow networks. CNN adopts a VGG-like (Simonyan & Zisserman, 2014) architecture with 14 layers, while ResNet is a modified version of ResNet18 (He et al., 2016) with 17 layers and incorporates shortcut connections. CNN and ResNet are specifically chosen to assess the efficacy of our method in training deep models.

**TinyCNN**

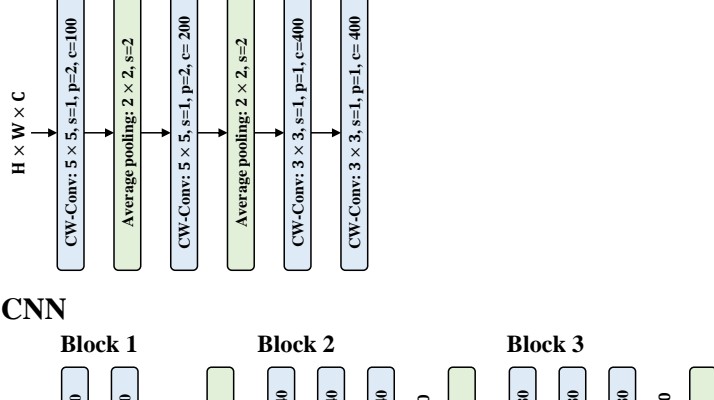

**CNN**

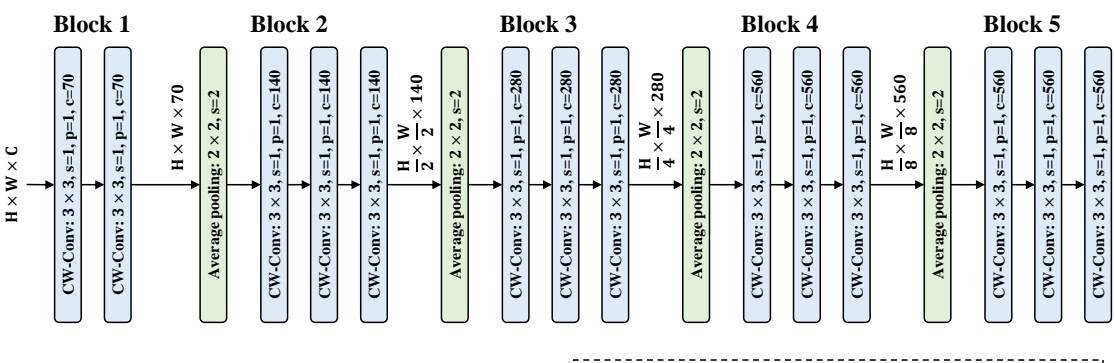

**ResNet**

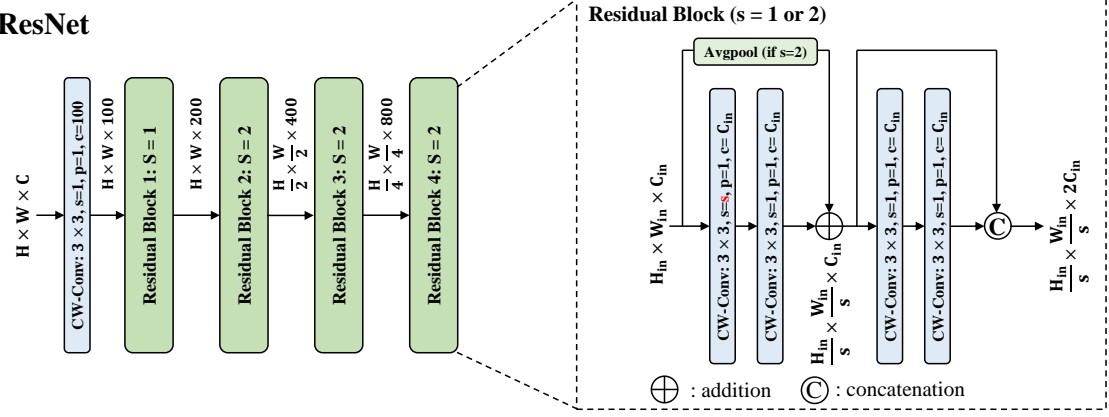

Figure 6: **The details of network architectures for the experiments.** (**Top**): TinyCNN is a 4-layer shallow CNN with a comparable number of parameters to other FF-related algorithms. (**Mid**): CNN represents a VGG-like architecture with 14 layers. (**Bottom**): Our ResNet architecture is modified from ResNet18, and its residual block structure is illustrated on the right.

## B  Hyperparameter Setting

In our method, we utilize Adam (Kingma & Ba, 2014) with a cosine annealing schedule (Loshchilov & Hutter, 2016), reducing the learning rate from 0.08 to 0.008, with a weight decay of 0.0005. Dropout with a rate of 20% is introduced after all hidden states. The models are initialized using He initialization (He et al., 2015). Training is conducted for 150 epochs with a batch size of 128 for all datasets. The experiment with original goodness in Table 4 uses the learning rate from 0.1 to 0.01 without dropout to have a better performance. The reproduced experiment of CwComp (Papachristodoulou et al., 2024) uses the same setting original paper.

## C  Data Augmentation Experiment

We conduct experiments on CIFAR10 with different augmentation settings based on the ResNet mentioned above architecture. We trained the model using the AdamW optimizer for 1000 epochs, starting with a learning rate of 0.08 and halving it every 100 epochs. The experiment without augmentation uses a dropout rate of 20%. There are two augmentation experiments, including standard augmentation and heavy augmentation. The standard augmentation contains random horizontal flips and random crops for a size of 32 with a padding of 4. The enhanced augmentation uses 4 kinds of augmentation techniques, including random horizontal flips, random rotation with a degree of 10, random scaling from 0.8 to 1.2, and color jittering by up to 0.2 shifting on brightness, contrast, saturation, and hue. Note that the experiments with augmentation do not use dropout, because, experimentally, the dropout causes the convergence too slow with data augmentation.

The training curve on the test set is depicted in Figure 7, and the final performance is listed in Table 6. As shown in Table 6, standard augmentation has no significant impact on performance, while enhanced augmentation improves performance from 87.16% to 88.72%. However, compared to the performance gains from data augmentation in traditional BP, the improvement with our method is less pronounced. Figure 7 exhibits that training without augmentation converges faster. At the 150th epoch, no augmentation outperforms standard and enhanced augmentation. This result demonstrates that DeeperForward is not sensitive to data augmentation.

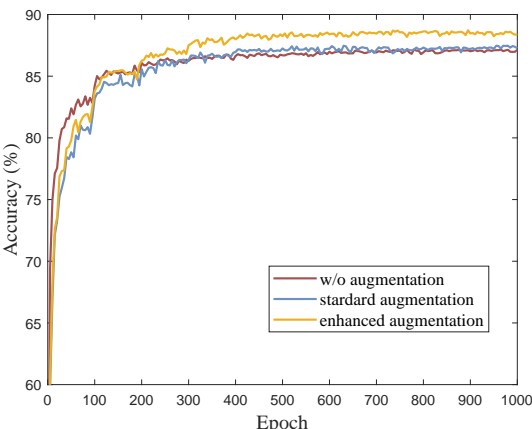

Figure 7: Comparison in test accuracy for data augmentation experiments on CIFAR10 for 1000 epochs. (**Without augmentation**): Training on original CIFAR10 with a dropout rate of 20%. (**Standard augmentation**): Standard data augmentation without dropout using random horizontal flips and random crops. (**Enhanced augmentation**): Enhanced data augmentation without dropout using random horizontal flips, random rotation of 10 degrees, random scaling from 0.8 to 1.2, and color jitter.

Table 6: Performances of different augmentation on CIFAR10.

|  | NO AUG. | STANDARD AUG. | ENHANCED AUG. |
|---|---|---|---|
| TEST ACC. | 87.16 | 87.47 | 88.72 |

## D  SCALING EXPERIMENT

To evaluate the performance of our method in deeper models, we tested it using deeper ResNet (He et al., 2016) models on CIFAR-10. We modified ResNet34 and ResNet101 by removing their fully connected layers, renamed as ResNet33 and ResNet100, and replacing the residual structures with those described in Figure 3. The first convolutional layer was adjusted to have a kernel size of 3 and 100 channels. ResNet33, like ResNet18, uses two kernel size 3 layers as the basic residual block, while ResNet100 employs a bottleneck residual block with three convolutional layers of kernel sizes 1, 3, and 1, where the third layer's output channels are four times the target number of channels. All models retain the original four stages, with corresponding channel numbers $\{100, 200, 400, 800\}$, each stage consisting of stacked residual blocks. In the ResNet33 structure, the four stages contain $\{3, 4, 6, 3\}$ blocks, while ResNet100 contains $\{3, 4, 23, 3\}$ blocks.

Table 7 presents the performance of various network depths on CIFAR-10, including the ResNet17 structure described in the main text. The experiments were conducted with the experimental setting mentioned in Section 4 and Appendix B and trained for 150 epochs on CIFAR-10. The results show that the ResNet17 model performs the best, while increasing the network depth to 33 layers did not yield better results. The ResNet100, which uses a bottleneck structure, performed worse. Tracking each layer's performance revealed significant drops after each $1 \times 1$ convolution layer, indicating that our method is unsuitable for the bottleneck structure. Additionally, the SIP layer retention results show that all layers were retained in ResNet33, whereas in ResNet100, only 32 layers were retained, with the latter layers being discarded due to poor performance.

Table 7: Performance on CIFAR10 for scaling experiments.

|  | RESNET17 | RESNET33 | RESNET100 |
|---|---|---|---|
| ACCURACY | **86.22** | 85.43 | 83.06 |

## E  EXPERIMENTS FOR DIFFERENT TYPES OF LAYERS

Our work uses channel-wise convolution (CW-Conv) for classification instead of fully connected (FC) layers. The experiments for different types of layers aim to compare the classification performance of different layer types, including a fully connected (FC) layer and two convolutional layers with different kernel sizes. The residual structure experiment's favorable results in deep layers of ResNet confirm the feasibility of classifying representations from these layers. Specifically, we replace the last 4 layers of ResNet with the layers to be tested and evaluate their local performance, as illustrated in Figure 8. The convolutional layer is the channel-wise convolution mentioned in the main body. The FC layer differs from the typical linear classifier, producing 1000 outputs divided equally into 10 groups for the 10 classes. Each class utilizes the mean of outputs from the corresponding group as its classification score. Using a general linear classifier with 10 outputs yields poor convergence.

Table 8 demonstrates that CW-Conv outperforms the FC layer significantly. Surprisingly, the experiment reveals that $3 \times 3$ kernels with fewer weights exhibit better performance. These findings suggest that the fully connected layer is superfluous in our method.

Table 8: Performances on different types of layers on CIFAR10.

|  | FC LAYER | CONV 5×5 | CONV 3×3 |
|---|---|---|---|
| ACCURACY | 76.92 | 82.85 | **83.25** |

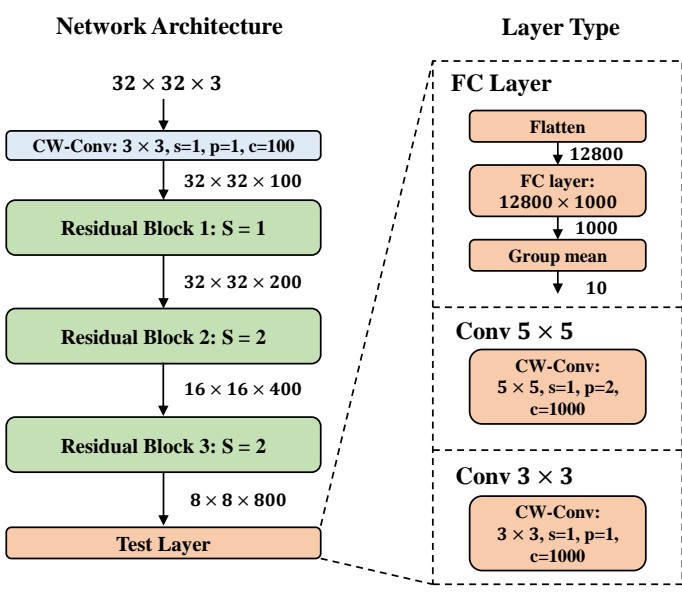

Figure 8: **The network architecture setting of layer type experiments.** (**Left**): The network architecture is modified by the ResNet model, replacing the last residual block with 3 types of layers. (**Right**): 3 types of layer architectures in this experiment, including an FC layer, a $3 \times 3$ channel-wise convolution, and a $5 \times 5$ channel-wise convolution.

## F    T-SNE Visualization

T-SNE (Van der Maaten & Hinton, 2008), short for t-distributed stochastic neighbor embedding, is a dimensionality reduction technique commonly used for visualizing high-dimensional data in lower-dimensional space. It emphasizes the preservation of local structure, making it effective for revealing underlying patterns and clusters within complex datasets.

In our experimental setup, we employed t-SNE to analyze the MNIST dataset. We selected t-SNE for its capability to capture intricate relationships between data points and represent them in a lower-dimensional space while preserving their local structures. The t-SNE analysis of the MNIST representation was validated using the hidden states of the 11th layer before group normalization in ResNet, which represents the last layer after the pruning operation conducted by the *signal integrating and pruning* (SIP) module.

The t-SNE visualization result is depicted in Figure 9. Before model training, the point clusters of each class in the MNIST dataset overlapped on boundaries. However, after training, a noticeable transformation occurred: the clusters became more distinctly separated, with discernible boundaries between them. This evolution underscores the model's ability to learn discriminative features and improve class separability.

Our utilization of t-SNE provided valuable insights into the dynamics of the MNIST dataset and highlighted the efficacy of our model in enhancing the separability of MNIST.

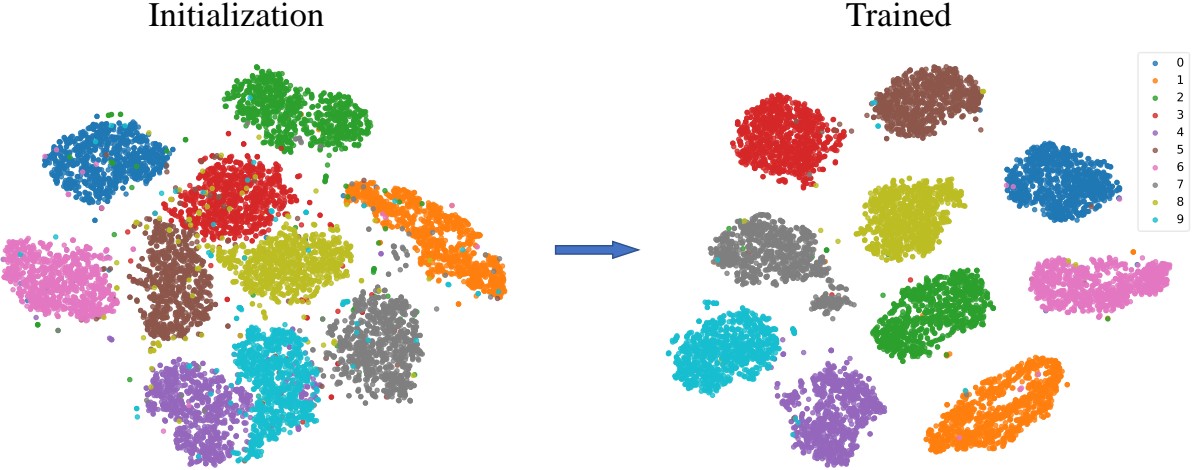

Figure 9: The t-SNE visualization of the MNIST representation at the hidden state of the 11th layer before group normalization in ResNet trained on MNIST, both before (initialization) and after training (trained). Each color corresponds to a different class.

## G  IMPLEMENTATION OF MODEL PARALLEL STRATEGY

Without the limitation of update locking, our method enables a model parallel strategy using pipeline programming. Pipeline programming is a parallel computing paradigm where complex tasks are divided into a series of independent and sequential stages. Each stage performs a specific task and passes its results to the next stage. This programming model resembles a factory assembly line, where each worker is responsible for completing a particular task in the process without waiting for the completion of the previous task. Tasks are divided into multiple stages that can execute simultaneously, thus speeding up the overall processing. By employing pipeline programming, we can enhance the efficiency and performance of computer systems, achieving faster data processing and reduced response times. The advantages of this programming model include parallelized processing, reduced latency, increased throughput, and better utilization of hardware resources.

Figure 10 illustrates our method's model parallel strategy implemented using pipeline programming based on multi-threading techniques. The network is divided into a series of independent and sequential threads. Each layer is organized into a thread to execute local optimization and generate data for the subsequent layer. As depicted in Figure 10, we employ first-in-first-out (FIFO) queues for communication among threads. The training set serves as the first FIFO designated for the initial layer. Each layer independently executes its specific task without waiting for the completion of other layers. This approach enables high parallelism, enhancing throughput, and optimizing hardware resource utilization. Algorithm 1 provides detailed insights into the implementation of each thread's functionality.

For clearer presentation, we simplify channel-wise convolution to the following equation:

$$(\mathbf{Z}, \hat{\mathbf{y}}) = \mathrm{CWConv}(\mathbf{X}; \boldsymbol{\theta}, G), \tag{14}$$

where $\hat{\mathbf{y}}$ and $\mathbf{Z}$ represent the classification output by the channel-wise convolution and the features extracted via layer normalization, respectively. $\mathbf{X}$ is the input, $\boldsymbol{\theta}$ denotes the convolutional kernel parameters, and $G$ is the number of classes.

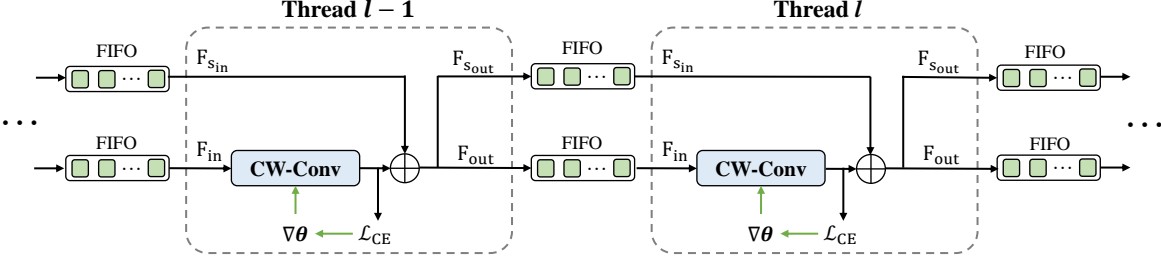

Figure 10: The implementation of the model parallel strategy of our method using pipeline programming based on the multi-threading technique. The network is segmented into a series of independent sequential threads, with each layer arranged within a thread to execute local optimization and produce data for the subsequent layer. Communication among threads is facilitated by first-in-first-out (FIFO) queues. It is important to note that we use the training set as the first FIFO, designated for the initial layer.

---

**Algorithm 1** A thread for local layer updates

---

1: **Input:** Input FIFO $F_{in}$, output FIFO $F_{out}$, FIFO for shortcut input $F_{sin}$, FIFO for shortcut output $F_{sout}$, parameters of convolutional kernel $\boldsymbol{\theta}$, number of class $G$, learning rate $\eta$, number of minibatch $N$.
2: **for** $i = 1, 2, ..., N$ **do**
3:     Wait for $F_{in}$ not empty
4:     $(\boldsymbol{X}, \boldsymbol{y}) \leftarrow F_{in}$     // pop data from $F_{in}$ into $(\boldsymbol{X}, \boldsymbol{y})$
5:     $(\boldsymbol{Z}, \hat{\boldsymbol{y}}) = \text{CWConv}(\boldsymbol{X}; \boldsymbol{\theta}, G)$
6:     // residual structure
7:     **if** residual structure is existed **then**
8:         Wait for $F_{sin}$ not empty
9:         $\boldsymbol{Z}_r \leftarrow F_{sin}$     // pop data from $F_{sin}$ into $\boldsymbol{Z}_r$
10:        **if** $\boldsymbol{Z}_r$ and $\boldsymbol{Z}$ differ in spatial dimension **then**
11:            $\boldsymbol{Z}_r = \text{AvgPool}(\boldsymbol{Z}_r)$     // downsampling $\boldsymbol{Z}_r$ to match $\boldsymbol{Z}$ in spatial dimension
12:        **end if**
13:        **if** concatenation type **then**
14:            $\boldsymbol{Z} = \text{Concat}(\boldsymbol{Z}, \boldsymbol{Z}_r)$
15:        **else**
16:            $\boldsymbol{Z} = \boldsymbol{Z} + \boldsymbol{Z}_r$
17:        **end if**
18:    **end if**
19:    **if** output connect to a shortcut **then**
20:        $F_{sout} \leftarrow \boldsymbol{Z}$     // push $\boldsymbol{Z}$ into $F_{sout}$
21:    **end if**
22:    $F_{out} \leftarrow (\boldsymbol{Z}, \boldsymbol{y})$     // push $(\boldsymbol{Z}, \boldsymbol{y})$ into $F_{out}$
23:    // update layer
24:    $\nabla \boldsymbol{\theta} = \nabla_{\boldsymbol{\theta}} \mathcal{L}_{CE}(\hat{\boldsymbol{y}}, \boldsymbol{y})$
25:    $\boldsymbol{\theta} = \boldsymbol{\theta} - \eta \cdot \nabla \boldsymbol{\theta}$
26: **end for**

---

## H    MEMORY-SAVING STRATEGY

Due to the update locking problems, BP requires a large amount of memory to store intermediate states throughout the process. Without this constraint, our method can employ a memory-saving strategy, efficiently releasing memory once the update for a layer is completed, as shown in Algorithm 2. In this strategy, the memory requirement is determined by the layer with the largest memory demand. This memory-saving strategy is particularly suitable for scenarios with constrained memory resources, such as edge computing.

---

**Algorithm 2** DeeperForward

---

1: **Input:** Dataset $\mathcal{B} \in \{\mathcal{B}_1, \mathcal{B}_2, ..., \mathcal{B}_N\}$, $\mathcal{B}_i = (\boldsymbol{X}_i, \boldsymbol{y}_i)$, number of layers $L$, number of classes $G$, learning rate $\eta$.
2: **for** $i = 1, 2, ..., N$ **do**
3: $\quad (\mathbf{Z}^{(0)}, \boldsymbol{y}) = \mathcal{B}_i$
4: $\quad$ **for** $l = 1, ..., L$ **do**
5: $\quad\quad (\mathbf{Z}^{(l)}, \hat{\boldsymbol{y}}^{(l)}) = \text{CWConv}(\mathbf{Z}^{(l-1)}; \boldsymbol{\theta}^{(l)}, G)$ $\qquad$ Eq. 14
6: $\quad\quad \nabla\boldsymbol{\theta}^{(l)} = \nabla_{\boldsymbol{\theta}^{(l)}} \mathcal{L}_{\text{CE}}(\hat{\boldsymbol{y}}^{(l)}, \boldsymbol{y})$ $\qquad$ Eq. 13
7: $\quad\quad \boldsymbol{\theta}^{(l)} = \boldsymbol{\theta}^{(l)} - \eta \cdot \nabla\boldsymbol{\theta}^{(l)}$
8: $\quad\quad$ Release memory of $\mathbf{Z}^{(l-1)}$, $\hat{\boldsymbol{y}}^{(l)}$ and $\nabla\boldsymbol{\theta}^{(l)}$
9: $\quad$ **end for**
10: **end for**

---

# I   DEACTIVATION NEURONS RATIO

To better explore the improvements of our method on the deactivated neurons issue, we conducted experiments and analysis on the deactivated neurons ratio across different layers. The experiments compared our combination of mean goodness and layer normalization with the original square goodness and normalization of the vector length, both tested on the same 17-layer ResNet structure in the CIFAR10 dataset. Both networks were trained with the same settings for 150 epochs on CIFAR10. The average deactivated neurons ratio for each layer was tested on the CIFAR10 test set, as shown in the Figure 11.

The results show that our mean goodness design indeed significantly reduces the deactivated neurons ratio. Specifically, in the shallow layers, square goodness deactivates a large number of neurons by focusing on outliers. Moreover, we observed that the mean goodness deactivated neurons ratio increases as the depth of the network increases. This phenomenon aligns with our observation that high-level feature matching becomes sparser in deeper layers, which also explains why deeper networks (such as ResNet33 and ResNet100 in Appendix D) do not show improved performance — extracting higher-level features from high-level features becomes increasingly difficult.

This experiment confirms that our method effectively addresses the deactivated neurons issue present in the original FF. Additionally, we further observe that high-level features extracted in deeper layers are relatively sparse, limiting the extraction of higher-level features. This results in deeper network layers failing to achieve optimal performance, which is a topic worthy of further exploration.

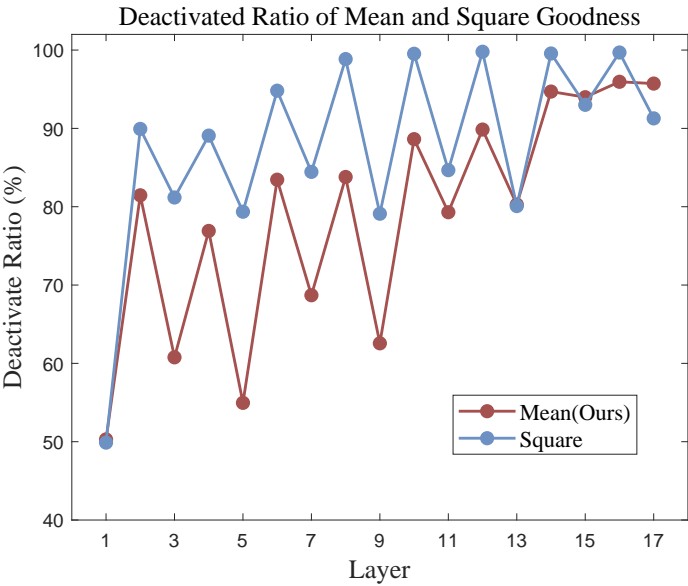

Figure 11: Comparison of mean goodness and square goodness in average deactivated neurons ratio per layer in 17-layer ResNet on CIFAR10's test set.

## J  IMPACT STATEMENTS

This paper presents work whose goal is to advance the field of Machine Learning. There are many potential societal consequences of our work, none which we feel must be specifically highlighted here.

