# OpenReview forum: "DeeperForward: Enhanced Forward-Forward Training for Deeper and Better Performance"
_ICLR.cc/2025/Conference — ICLR 2025 Poster_

### Official Review · Reviewer_eCNM · 2024-10-17

**Soundness:** 3
**Presentation:** 2
**Contribution:** 3
**Rating:** 6
**Confidence:** 4

**Summary:**

This paper presents the DeeperForward algorithm, extending the Forward-Forward approach to deeper networks. The paper introduces a goodness design, combining mean goodness and layer normalization, which partially addresses issues in the effective training of deep networks: feature scaling, redundant normalization, and deactivated neurons. It’s a bit innovative but not very clear in expression.

**Strengths:**

- The article proposes an innovative design based on the original FF method to solve the inherent problems of BP and FF.
- From the experimental results, it can be seen that DeeperForward has good performance.

**Weaknesses:**

- The abstract lacks a brief description of the new parallel strategy.
- Figure 1 is confusing. Firstly, it would be more appropriate to compare the performance gap between your method and SOTA in Figure 1. Secondly, your method only has a single forward propagation path, which is inconsistent with the enhanced FF in your title.
- ‘However, this approach fails to decouple goodness and leaks goodness to the next layer, hindering deeper layers from learning new features and causing overfitting.’ Authors have repeatedly used this general and assertive conclusion in the introduction to evaluate the drawbacks of other methods, but I have not seen a specific reason. This is not very detailed and almost unconvincing. Moreover, it is evident that this conclusion has nothing to do with feature scaling.
- ‘Although CwComp (Papachristodoulou et al., 2024) improves performance using squared goodness and batch normalization, it leaks goodness information, leading to overfitting in deeper networks.’ According to your description, the omission of goodness of fit information directly leads to overfitting, which is clearly not convincing. Any defects in the model can lead to common problems of overfitting. There is a lack of deeper reasons between the two.
- ‘We adopt layer normalization for better feature scaling with identical mean and standard deviations, instead of L2 normalization.’ You have expressed the drawbacks of L2 normalization and proposed so-called alternative methods. But why do we need to use your method to replace it? What is the motivation? You cannot simply express that the original method has shortcomings but your method has advantages, and then your method can become an alternative solution without deeper motivation and logic.
- The format of subheadings under section 3.2 is inconsistent.
- From Sec4.1, it can be seen that GPU memory is sufficient, but there is a lack of experiments on large datasets such as ImageNet.
- 8.Sec4.5 only compared GPU and speed with BP, but according to what you described earlier, you have made improvements based on FF. There is a lack of results compared to FF here.

**Questions:**

Q1. ‘This approach addresses the bio-plausibility issues of backpropagation and overcomes the limitations of FF concerning model size.’ In the Method section, the problem solved by this method is inconsistent with the problem you mentioned in the introduction that needs to be solved. Does model size refer to feature scaling? What is the relationship between the bio funding issues of backpropagation and deactivated neurons?

Q2. ‘The pruning process aims to select an optimal combination of layers for refining the network and eliminating redundant layers.’ How is it selected and what are the criteria for selecting combinations?

Q3. Where was Xr extracted from in Figure 3? Any layer? Because deeper layers contain higher semantic information, where Xr comes from is a critical issue for performance.

Q4. The DeeperForward you proposed is similar in principle to the methods[1,2] in the Local Learning field. What are your strengths?

[1] Xiong, Yuwen, Mengye Ren, and Raquel Urtasun. "Loco: Local contrastive representation learning." Advances in neural information processing systems 33 (2020): 11142-11153.

[2] Su, Junhao, et al. "HPFF: Hierarchical Locally Supervised Learning with Patch Feature Fusion." arXiv preprint arXiv:2407.05638 (2024).

If the author resolves my questions, I will consider raising my rating.

---

> ### Author Response · Authors · 2024-11-22
> **Response to Reviewer 4 [Part 1/2]**
>
> Thank you for your valuable and detailed review. **The latest revised version has been modified and uploaded based on the reviewers' comments.**
>
> **Response to concerns in weakness:**
>
> 1.  Thank you for your suggestions and corrections regarding the writing of our paper. We have incorporated the necessary changes in the revised version as follows:  **(1) a brief description of model parallel strategy in abstract. (2)  The format in Section 3.2 has been adjusted to ensure consistency**.
>
> 2. Figure 1 in our paper primarily highlights the distinction between our approach and other bio-plausible training methods. **In bio-plausible training research, the differences in training frameworks are a key characteristic, not just the performance disparities.**   To provide a more comprehensive comparison, we include comparisons with other bio-plausible methods in **Figure 1**, not just FF methods.  **Although we use a single forward pass, we still adhere to the FF training framework, and the use of a single forward pass is more efficient.**
>
> 3. **The decoupling of goodness**: The FF method uses goodness as a classification metric at each layer, as seen in the original FF, which uses the sum of squared outputs as goodness. If the output is directly used as the input for the next layer, it retains the classification information from the previous layer. For instance, in the case of a positive class, the input elements are relatively large, and for other classes, they are smaller. **This allows the next layer to make new predictions based on the predictions carried over from the previous layer (e.g. an identity transformation), without the need for further image feature learning.** Therefore, FF proposes the decoupling of goodness and uses L2 norm to normalize the input goodness information to a consistent value of 1. The issue in the **CwComp** paper is that **they did not decouple goodness**, allowing subsequent layers to rely on classification information from the previous layers, which leads to overfitting in deep networks.
>
> 	**While L2 norm can decouple goodness, it cannot control the input within a reasonable range of mean and variance**, which introduces **Feature Scaling** issues and lowers performance. To address this, we propose using **Layer Normalization** and **Mean Goodness**, which simultaneously solve the problems of decoupling goodness and feature scaling.
>
>
>
> 4.  As explained in the **Introduction**, we use **Mean Goodness** and **LayerNorm** as alternatives to address the challenges in FF design, specifically the issues of **feature scaling** and **deactivated neurons**. The **square goodness** focuses more on the few outlier outputs, which leads to many outputs being deactivated, resulting in lack of features for the subsequent layers. Our **mean goodness** provides a more balanced focus on all elements, ensuring that more neurons are activated.   Furthermore, as mentioned in the **Method** section, in the case of **square goodness**, the weights corresponding to the deactivated neurons are not updated. However, with **mean goodness**, the weights of all neurons can still be updated.
>
> 	**To provide a more comprehensive explanation of this motivation**, we have added **Appendix I** to compare the deactivated neurons in both cases and to confirm that our method results in more activated neurons.
>
>
>
> 5. As mentioned in the **Conclusion** under **Limitations**, tasks with large-scale datasets like ImageNet, which involve 1000 classes, remain a challenge for our method. **This is because the number of channels per layer must be a multiple of the number of classes, and this multiplier is preferably greater than 10 to extract sufficient information. This requirement leads to impractical memory and time consumption for real-world applications.** We experimented with a ResNet17 architecture where the channel numbers for the 4 stages were modified to [1000, 1000, 2000, 2000] and trained for 20 epochs. However, the Top-5 accuracy was below 10%, due to the inadequacy of channels.
>
> 	**Currently, pure FF-based methods are predominantly evaluated on benchmarks like CIFAR10, MNIST, and FMNIST to validate feasibility, making our comparisons fair and reasonable.** Moreover, our method significantly improves the performance of FF-based approaches on these benchmarks. Nonetheless, applying FF-based methods to larger datasets remains an important direction for future exploration.
>
>
>
> 6.  The **model parallel strategy** is primarily compared with BP's **data parallel** method to evaluate our parallel performance. Other FF-based methods are typically limited to small networks (within 4 layers) and rely on **progressive learning**, training layer by layer rather than the entire network simultaneously. **These methods lack parallel acceleration strategies and are therefore not directly comparable.** As a result, we focus on comparing our approach with the commonly used **DDP** method on BP.

---

> ### Author Response · Authors · 2024-11-22
> **Response to Reviewer 4 [Part 2/2]**
>
> **Response to concerns in question:**
>
> 1. **( We have added experiments on deactivated neurons in Appendix I to make the paper more convincing. )** The term "model size" refers to the fact that prior FF methods have only been implemented in shallow networks. The issues of bio-plausibility in BP and model size in FF are distinct and address different concerns.
>
> + First, **bio-plausibility** is a challenge associated with backpropagation, as described in **lines 27–37** of the introduction. These challenges include weight transport, non-local updates, freezing activity, and update locking. Our method addresses these issues by leveraging the characteristics of the FF framework. Specifically, by setting independent objectives for each layer, we eliminate the need for error backpropagation, thereby resolving bio-plausibility concerns.
> + Second, the **limitations of FF concerning model size** refer to the observation that current FF-based methods are only effective in shallow networks. Deeper networks fail to improve performance or suffer from overfitting. In the introduction, we identified two main causes for this limitation: **Feature Scaling** and **Deactivated Neurons**.
>   - **Feature Scaling**: The original FF approach relies on L2 normalization, which cannot adequately control feature values within a reasonable mean and variance range. This results in large feature discrepancies, leading to overfitting. In this work, we address this issue by designing the network with **Layer Normalization**.
>   - **Deactivated Neurons**: The original FF method uses square goodness, which disproportionately emphasizes large outliers, resulting in a significant number of deactivated neurons. We resolve this problem by introducing **mean goodness**, which provides a more balanced approach. **In the new vision of the paper, we have added the experiment on deactivated neurons on Appendix I.**
>
> These improvements allow our method to effectively scale to deeper networks while addressing the bio-plausibility challenges inherent to backpropagation.
>
>
>
> 2. **( We have rewritten the SIP part to make it clearer. )** In other FF-based methods, it is common practice to simply sum the goodness values of the last one or several layers as the final output.   Our SIP strategy, however, seeks the optimal combination of layers whose cumulative goodness achieves the best results.   **Specifically, we evaluate various combinations by calculating accuracy on untrained data and selecting the best-performing combination as our final choice.** Given L layers, there are 2^L possible combinations, making an exhaustive search computationally prohibitive. To reduce complexity, **we restrict the combinations to only contiguous layers—summing the goodness values from a specified start layer to an end layer—thus reducing the search space to (L+1)L/2 combinations.**
>
>
>
> 3. **Xr serves as the input to the shortcut in the residual structure and is also the output from a preceding layer.**  Figure 3 illustrates the structure of a residual connection, where the output features are fused with the output Xr from a previous layer to generate new features.  **Xr can be the output of any preceding layer, depending on the network design.** As stated in the paper, we adopted the residual connection strategy based on the ResNet18 architecture. The detailed design of the entire network model is provided in Appendix A.
>
>
>
> 4.  **(The results of HPFF has been added in Table1.)** Thank you for providing the two papers: **Loco** and **HPFF**. Both papers incorporate auxiliary networks in the layers to perform block-wise local backpropagation training. **Loco** is based on contrastive learning, while **HPFF** uses supervised learning. **Both methods still rely on backpropagation's powerful learning capabilities, but they face issues related to bio-plausibility, such as the fact that neurons do not perform both forward and backward transmissions simultaneously, as in the brain, but rather in a one-directional manner.**
>
> 	The **Forward-Forward (FF)** method, which is brain-inspired, aims to replace backpropagation by exploring a training approach that more closely aligns with the mechanisms of the brain, specifically by updating weights during forward computation. In our method, we compute the weight updates simultaneously with the output at each layer, following the FF approach **without relying on auxiliary networks**.   While the performance still lags behind BP-based methods to some extent, **our approach significantly enhances the capability of FF-based methods, bringing us closer to a more brain-like training process**.
>
> Thank you for the insightful recommendation!

---

> > ### Comment · Reviewer_eCNM · 2024-11-24
> >
> > Your reply basically answered my questions and briefly explained the limitations of your research. I will improve my rating.

---

### Official Review · Reviewer_YDPM · 2024-10-21

**Soundness:** 3
**Presentation:** 3
**Contribution:** 2
**Rating:** 6
**Confidence:** 5

**Summary:**

The author proposed a novel approach, DeeperForward, to enhance the Forward-Forward algorithm. This paper introduced a goodness design based on layer norm and mean goodness to solve issues like feature scaling and deactivated neurons. This paper also demonstrates the effectiveness of DeeperForward using 17-layer CNN on various datasets. Additionally, this paper states a model-parallel strategy to reduce training time.

**Strengths:**

1. The proposed modifications, especially mean goodness and layer normalization, effectively address the limitations of the FF algorithm in deeper architectures.

2. The model-parallel strategy greatly enhances the training efficiency on multi-GPU setups, enabling the method to scale well across deeper networks.

**Weaknesses:**

1. **Dataset Size Limitations**: The reviewer acknowledges that DeeperForward has primarily been tested on relatively small datasets, such as CIFAR-10 and MNIST. While these datasets are widely used benchmarks, the reviewer agrees that evaluating DeeperForward on larger datasets, such as ImageNet, would provide more insight into its scalability and effectiveness. The reviewer recommends conducting experiments on larger datasets to assess their performance under different conditions thoroughly.

2. **Choice of Architecture Depth**: The reviewer notes that the 17-layer CNN was selected as a balance between computational efficiency and depth sufficient to highlight the advantages of DeeperForward over existing methods in moderately deep networks. However, the reviewer wonders if testing DeeperForward on even deeper architectures (e.g., 50 or 100 layers) would provide valuable insights. The reviewer encourages extending the evaluation to larger models to assess its performance and generalization capabilities on very deep networks.

3. **Accuracy and Comparison with Block-wise Learning**: The reviewer points out the lack of significant accuracy improvement in Table 1 compared to non-FF and other FF methods. While the method focuses on improving training efficiency and enabling deeper networks, the reviewer highlights recent works in block-wise learning[1,2] that have shown greater improvements in accuracy, surpassing backpropagation by over 5%. The reviewer is curious if DeeperForward could incorporate block-wise learning concepts to improve accuracy while retaining the benefits of the forward-forward training scheme.

4. **Scalability on Multiple GPUs**: The reviewer observes that the speedup ratios in Table 5 reflect performance gains using one and two GPUs but would like to see tests on more GPUs to provide a more comprehensive view of DeeperForward's scalability. The reviewer suggests extending parallelization experiments to include setups with more GPUs to provide a broader evaluation of how the method scales with additional hardware resources.

5. **Training Curve Convergence**: The reviewer notes concern regarding the convergence behavior shown in Figure 7. The training curve does not fully converge within the current number of epochs, and the reviewer would like to see further experiments to determine the number of epochs required for stable convergence. The reviewer recommends updating the results with extended training to provide a clearer picture of when and how the method reaches its optimal performance.

[1] Cheng, Anzhe, et al. "Unlocking Deep Learning: A BP-Free Approach for Parallel Block-Wise Training of Neural Networks." ICASSP 2024-2024 IEEE International Conference on Acoustics, Speech and Signal Processing (ICASSP). IEEE, 2024.

[2] Pyeon, M., Moon, J., Hahn, T. and Kim, G., 2020. Sedona: Search for decoupled neural networks toward greedy block-wise learning, in International Conference on Learning Representations.

**Questions:**

See weakness above

---

> ### Author Response · Authors · 2024-11-22
> **Response to Reviewer 3 [Part 1/2]**
>
> Thank you for your valuable and detailed review. **The latest revised version has been modified and uploaded based on the reviewers' comments.**
>
> 1. As mentioned in the **Conclusion** under **Limitations**, tasks with large-scale datasets like ImageNet, which involve 1000 classes, remain a challenge for our method. **This is because the number of channels per layer must be a multiple of the number of classes, and this multiplier is preferably greater than 10 to extract sufficient information. This requirement leads to impractical memory and time consumption for real-world applications.** We experimented with a ResNet17 architecture where the channel numbers for the 4 stages were modified to [1000, 1000, 2000, 2000] and trained for 20 epochs. However, the Top-5 accuracy was below 10%, due to the inadequacy of channels.
>
> 	**Currently, pure FF-based methods are predominantly evaluated on benchmarks like CIFAR10, MNIST, and FMNIST to validate feasibility, making our comparisons fair and reasonable.** Moreover, our method significantly improves the performance of FF-based approaches on these benchmarks. Nonetheless, applying FF-based methods to larger datasets remains an important direction for future exploration.
>
> 2. **( We have added the experiment on deactivated neurons in Appendix I to analyze the relationship between the deactivated neuron ratio and the number of layers. )** As discussed in **Appendix D** of the paper, we conducted experiments on architecture depth, including **ResNet models with 33 and 100 layers**. The results show that both the 33-layer and 100-layer ResNet models perform worse than the 17-layer model.
>
> 	| ResNet17 | ResNet33 | ResNet100 |
> 	| - | - | - |
> 	| 86.22    | 85.43    | 83.06     |
>
>
> 	Upon analysis, we found that the 33-layer model suffers from slower channel expansion as the depth increases, leading to weaker feature extraction capabilities in the shallow layers compared to the 17-layer model. On the other hand, the **100-layer model uses a bottleneck structure**, which limits the local feature extraction capacity and makes it less compatible with our method. Additionally, the FF algorithm is inherently greedy, aiming to extract higher-level features from the previous layer. However, **it becomes more difficult to extract even higher-level features from already high-level features in deeper layers**.
>
> 	In deeper networks, convolutional layers focus on matching high-level features, which results in only **a small number of neurons being activated**. This, in turn, makes it more challenging for subsequent layers to learn new information. To illustrate this, we have added **Appendix I**, which provides a detailed analysis of the relationship between the deactivated neuron ratio and the number of layers.
>
> 	In summary, the 17-layer architecture delivers better performance, and our method marks a significant improvement over other FF-based approaches.
>
> 3. **(The results of mentioned papers have been added in Table 1.)** Thank you for providing the two block-wise local update papers. **Both leverage auxiliary networks to perform local backpropagation updates and even surpass BP in performance.** However, FF research focuses on brain-inspired training, aiming to address BP's bio-implausibility issues, such as the one-directional nature of neurons. In FF, weights are updated during the forward process, eliminating the need for distinct forward and backward phases.   In our original paper, non-FF methods were categorized as brain-inspired algorithms. **We have added the papers you provided to Table 1 for a more comprehensive comparison.** Furthermore, combining FF algorithms with block-wise BP represents a promising research direction that warrants further exploration.
> 4. **(Section 4.5 has been modified based on your suggestion.)** Thanks for your suggestion. To test the performance on more GPUs, we used another machine with **4 GPUs (TITAN X)**. The test results are as follows:
>
> 	| METHOD   | 1 GPU             | 2 GPUs             | 4 GPUs             |
> 	| -------- | -- | - | - |
> 	| **BP (DDP)** | 51.98s (1.0x)     | 32.70s (1.59x)     | 19.92s (**2.61x**) |
> 	| **Ours** | **36.38s**(1.0x) | **20.77s (1.75x)** | **14.68s** (2.48x) |
>
>
> 	In the extended experiments, we observed that the acceleration rate on 4 GPUs was inferior to DDP. **By monitoring GPU utilization, we found that GPU usage initially reached 100% at the start of training but subsequently dropped to around 50% on each GPU. This issue arises due to varying training speeds across layers, causing queue saturation and resulting in bottlenecks.** Pipeline programs can only achieve their maximum advantage when computational loads across layers are balanced. In this paper, we demonstrate the feasibility and potential of model parallelism using a simple multi-threading approach. Further optimization can involve integrating advanced pipeline programming techniques, which would be a promising direction for future research.

---

> ### Author Response · Authors · 2024-11-22
> **Response to Reviewer 3 [Part 2/2]**
>
> 5. **( We have modified Appendix C as your suggestion. )** Thanks for your suggestion. We have revised our experimental setup to achieve better convergence in **Appendix C**. We used **1000 epochs** and applied **StepLR to reduce the learning rate by half every 100 epochs**, with an initial learning rate of **0.08**. The final results on CIFAR10 are as follows:
>
> 	| no augmentation | 2 augmentations | 4 augmentations |
> 	| --------------- | --------------- | --------------- |
> 	| 87.16           | 87.47           | 88.72           |
>
>
> 	**As shown in the table, standard augmentation has no significant impact on performance, while enhanced augmentation improves performance from 87.16% to 88.72%. However, compared to the performance gains from data augmentation in traditional BP, the improvement with our method is less pronounced.** This result demonstrates that DeeperForward is not sensitive to data augmentation, compared to BP.
>
> Thank you for the insightful recommendation!

---

> > ### Comment · Reviewer_YDPM · 2024-11-24
> >
> > Thank you for your detailed and thoughtful response. I appreciate the effort you have put into addressing the concerns raised in the review and the additional experiments and analyses you conducted. Your revisions and added analyses significantly strengthen the manuscript. I do not have additional concerns at this stage, and I believe your paper makes an important contribution to the field of FF-based methods. I will raise the score for your updated manuscript.

---

### Official Review · Reviewer_GH5m · 2024-11-04

**Soundness:** 3
**Presentation:** 3
**Contribution:** 2
**Rating:** 5
**Confidence:** 3

**Summary:**

This work aims to improve forward-forward training and enable deeper networks by providing layer normalization and mean goodness to overcome certain limitations.  Experiments show that the performance can be improved even with 17-layer CNNs.  Datasets like CIFAR-10, MNIST, and Fashion-MNIST are involved in illustrating the benefits of the proposed DeeperForward method.  In addition, a model parallel strategy and memory saving strategy are proposed to enhance training efficiency and costs.

**Strengths:**

The brain-inspired forward-forward approach is certainly an interesting and novel area to explore. It’s great to see efforts advancing in this direction.

The writing and presentation are easy to follow.

The experiments in the study are substantial.

**Weaknesses:**

The main issue of the study is the performance of the proposed DeeperForward method.   Compared with sota models using BP, DeeperForward falls behind on Fashion MNIST and has an even bigger gap on CIFAR-10.  Its accuracy on MNIST is marginally comparable to other methods. However, that is not particularly strong considering the simplicity of MNIST.  The results suggest the exploration of FF is ongoing and still faces some challenges.

Line 309, "we employ a combination strategy for selecting consecutive layers"  It is not clear what the strategy is and how this strategy reduces the complexity from O(2^L) to O(L^2).

It is not clear how the number of layers would impact the performance.  Is 17 the top limit? Would an architecture larger than 17 layers suffer from some kind of vanishing gradient problem?

Despite some discussions in Section 4.5, the computational cost aspect of DeeperForward is inadequate. What is the total GPU time needed to complete a task, for example, CIFAR-10 classification?  That measure would allow comparison with other non-FF methods more comprehensively. This should be a critical part of this study.

The memory-saving strategy lacks details.  Appendix H does not provide any measures or evidence on how this strategy could reduce memory usage for DeeperForward.

Similarly, the parallel strategy needs more details as well.  What if the number of GPUs is more than 2?  Also, what would happen if the number of GPUs is greater than the number of layers?  Would the extra GPUs stay idle? Fig 4 does not provide such insight.

**Questions:**

See above as the questions are mainly addressing weaknesses.

---

> ### Author Response · Authors · 2024-11-22
> **Response to Reviewer 2 [Part 1/2]**
>
> Thank you for your valuable and detailed review. **The latest revised version has been modified and uploaded based on the reviewers' comments.**
>
> 1.  As mentioned in the introduction, BP methods suffer from bio-implausibility issues. Therefore, the research on Forward-Forward (FF) training primarily focuses on finding suitable bio-plausible methods to replace BP. Bio-plausible training updates weights during the forward computation, without relying on error backpropagation, which aligns with the unidirectional nature of neurons. Currently, bio-plausible training lags behind BP in terms of performance, including FF-based methods.  Our research has identified several reasons for the underperformance of FF methods and has addressed them. **Compared to other FF-based approaches, we have achieved a significant performance improvement, narrowing the gap with BP and highlighting the potential of FF-based methods.**
>
>
>
> 2. **( We have rewritten the SIP part to make it clearer. )** In other FF-based methods, it is common practice to simply sum the goodness values of the last one or several layers as the final output.   Our SIP strategy, however, seeks the optimal combination of layers whose cumulative goodness achieves the best results.   Specifically, we evaluate various combinations by calculating accuracy on untrained data and selecting the best-performing combination as our final choice.  **Given L layers, there are 2^L possible combinations, making an exhaustive search computationally prohibitive.** To reduce complexity, **we restrict the combinations to only contiguous layers—summing the goodness values from a specified start layer to an end layer—thus reducing the search space to (L+1)L/2 combinations.**
>
>
>
> 3.  **( We have added the experiment on deactivated neurons in Appendix I to analyze the relationship between the deactivated neuron ratio and the number of layers. ) **The study on **deeper architectures** is discussed in detail in **Appendix D**, where we analyze and explore the performance of** ResNet33 and ResNet100**. These architectures are derived from ResNet34 and ResNet101, respectively, by removing the fully connected layers and adjusting the number of channels. The following table presents the performance comparison:
>
> 	|              | ResNet17 | ResNet33 | ResNet100 |
> 	| ------------ | -------- | -------- | --------- |
> 	| **CIFAR10 Acc.** | 86.22    | 85.43    | 83.06     |
>
> 	The results show that the 17-layer model performs the best. We attribute this to several factors. Unlike BP, FF can only perceive inputs from the previous layer and greedily outputs higher-level features. In deeper networks, convolutional layers focus on matching high-level features, which results in only **a small number of neurons being activated**. This, in turn, makes it more challenging for subsequent layers to learn new information. To illustrate this, we have added **Appendix I**, which provides a detailed **analysis of the relationship between the deactivated neuron ratio and the number of layers**.   Additionally, **Section 4.2** and **Table 2** demonstrate that the larger the number of channels, the better the feature extraction performance. In the ResNet100 model, the bottleneck structure significantly reduces the number of channels to minimize parameters, which in turn degrades performance.  **Our method does not rely on error propagation, so it does not encounter the vanishing gradient problem.**
>
>
>
> 4. This experiment primarily aims to demonstrate the parallel acceleration performance of the **model parallel strategy**, comparing it with the commonly used **distributed data-parallel (DDP)** method. Although other non-FF methods adopt different training frameworks, they all rely on DDP for multi-GPU parallel training. Therefore, we use BP as the baseline for comparing the parallel acceleration of the **model parallel strategy** and **DDP**.
>
> 	In this study, we measured the computational cost of **DeeperForward**, particularly its performance on the CIFAR-10 classification task. On **2 GPUs (TITAN X 24GB)**, DeeperForward completed 150 epochs of training in **51 minutes 55 seconds**, achieving a speedup of **1.75x** compared to the single-GPU case. In contrast, BP using DDP completed 150 epochs of training in **81 minutes 45 seconds**, achieving a speedup of **1.59x** compared to the single-GPU case.

---

> ### Author Response · Authors · 2024-11-22
> **Response to Reviewer 2 [Part 2/2]**
>
> 5. **( We have rewritten the Memory-saving strategy part to make it clearer. )** The **memory-saving strategy** is a layer-by-layer update strategy, which consists of the following steps: **(1) Perform computation for the current layer. (2) Update weights in the current layer. (3) Pass the output to the next layer. (4) Release all intermediate memory used by the current layer. (5) Repeat (1)-(4) to update subsequent layers.** Since there is no limitation from freezing activities, this strategy achieves memory savings by promptly releasing memory after each layer's computation, thus minimizing memory usage during training.
>
> 6. Figure 4 illustrates our **model parallel strategy**, where different layers are assigned to specific GPUs (the same GPU can host multiple layers) and parallel training is performed using a pipeline program. In the pipeline program, each layer does not need to wait for the entire network to complete an update before proceeding with the next round of training. For example, in our experiments, we assigned layers 1-8 to GPU 0 and the remaining layers to GPU 1 for training. **When using 4 GPUs, we can allocate 4-5 layers to each GPU to balance the computational load. If the number of GPUs exceeds the number of layers, we can assign one GPU per layer, leaving the extra GPUs unused.** This approach helps to optimize GPU utilization and alleviate the computational burden on each GPU.
>
> Thank you for the insightful recommendation!

---

> > ### Comment · Reviewer_GH5m · 2024-11-23
> >
> > Thanks for the response.  I have raised my score.  The key concern regarding DeeperForward's performance still remains.  Bio-implausibility is certainly an issue worthy to be explored.  Its priority would not be higher than that of accuracy.  After all, we are not pursuing bio plausible flapping wing aircraft, aren't we?

---

### Official Review · Reviewer_occo · 2024-11-12

**Soundness:** 3
**Presentation:** 4
**Contribution:** 3
**Rating:** 8
**Confidence:** 2

**Summary:**

The current paper proposes a novel design of bio-logically plausible algorithms utilizing layer normalization and mean goodness, which extends the Forward-Forward to non-trivial networks. Extensive experiments across datasets and models validate the approach.

**Strengths:**

The paper enables training conventional models with the FF algorithm.

Diagrams are well designed, clear, and nice.

The explanation of the method is clear.

This approach scales well to multiple GPUs

I like the well written background section.

**Weaknesses:**

I would like to see experiments on Imagenet or other large scale datasets.

I'm interested in an experiment that shows that the portion of deactivated neurons is reduced.

**Questions:**

Can you show that the portion of deactivate neurons is reduced?

---

> ### Author Response · Authors · 2024-11-22
> **Response to Reviewer 1**
>
> Thank you for your valuable and detailed review. **The latest revised version has been modified and uploaded based on the reviewers' comments.**
>
> 1. As mentioned in the **Conclusion** under **Limitations**, tasks with large-scale datasets like ImageNet, which involve 1000 classes, remain a challenge for our method. **This is because the number of channels per layer must be a multiple of the number of classes, and this multiplier is preferably greater than 10 to extract sufficient information. This requirement leads to impractical memory and time consumption for real-world applications.** We experimented with a ResNet17 architecture where the channel numbers for the 4 stages were modified to [1000, 1000, 2000, 2000] and trained for 20 epochs. However, the Top-5 accuracy was below 10%, due to the inadequency of channels.
>
>  	**Currently, pure FF-based methods are predominantly evaluated on benchmarks like CIFAR10, MNIST, and FMNIST to validate feasibility, making our comparisons fair and reasonable.** Moreover, our method significantly improves the performance of FF-based approaches on these benchmarks. Nonetheless, applying FF-based methods to larger datasets remains an important direction for future exploration.
>
>
>
> 2.  **( We have added the experiment on deactivated neurons in Appendix I to make the paper more convincing. )** Thank you for your feedback, which has significantly enhanced the comprehensiveness of our paper. We tested the average **deactivated neurons ratio** for each layer in the trained 17-layer ResNet on the CIFAR-10 test set, comparing our **mean goodness** design with the original **square goodness** design. The results are presented in the following table:
>
>  	The results show that our **mean goodness** design indeed significantly reduces the deactivated neurons ratio. Specifically, in the shallow layers, **square goodness** deactivates a large number of neurons by focusing on outliers. Moreover, we observed that the **mean goodness** deactivated neurons ratio increases as the depth of the network increases. This phenomenon aligns with our observation that high-level feature matching becomes sparser in deeper layers, which also explains why deeper networks (such as ResNet33 and ResNet100) do not show improved performance — extracting higher-level features from high-level features becomes increasingly difficult.
>
>  	**Your suggestion has been critical for broadening the analysis in our paper, and we have added a new section in Appendix I to describe and analyze the effect of the deactivated neurons ratio.**
>
> Table: Deactivated neurons ratio (%) for each layer in 17-layer ResNet on CIFAR10.
>
> | layer  | 1         | 2         | 3         | 4         | 5         | 6         | 7         | 8         | 9         |
> | ------ | --------- | --------- | --------- | --------- | --------- | --------- | --------- | --------- | --------- |
> | square | **49.75** | 99.17     | 91.02     | 99.48     | 91.67     | 99.28     | 83.02     | 99.41     | 80.14     |
> | mean   | 50.29     | **81.46** | **60.77** | **76.91** | **54.96** | **83.45** | **68.70** | **83.80** | **62.56** |
>
>
> | layer  | 10        | 11        | 12        | 13        | 14        | 15        | 16        | 17        |
> | ------ | --------- | --------- | --------- | --------- | --------- | --------- | --------- | --------- |
> | square | 99.45     | 87.06     | 98.91     | 90.38     | 98.39     | 96.00     | 99.33     | **94.89** |
> | mean   | **88.63** | **79.29** | **89.85** | **80.27** | **94.70** | **93.98** | **95.95** | 95.72     |
>
>
>
> Thank you for the insightful recommendation!

---

### Author Response · Authors · 2024-11-22
**The revised draft is now available.**

We sincerely appreciate the reviewers' professional and insightful feedback, which has significantly improved the quality and clarity of our paper. We have made the following revisions, **highlighted in blue in the revised manuscript**, to provide a more comprehensive analysis and strengthen the arguments:

1. **Abstract**: Added a brief introduction to the _model parallel strategy_ to outline its importance and relevance.
2. **Section 3.2**: Unified the formatting of subheadings for consistency and readability.
3. **Signal Integrating and Pruning Module (Section 3.2)**: Revised the description of the SIP module to clarify the details of selection and pruning, making the explanation more precise.
4. **Memory-Saving Strategy (Section 3.3)**: Rephrased and elaborated on the memory-saving strategy, including additional step-by-step details for improved clarity.
5. **Table 1**: Expanded the comparison with additional block-wise local BP methods (e.g., HPFF, SEDONA, BWBPF) to provide a more comprehensive evaluation against various method categories.
6. **Section 4.5**: Added a new analysis comparing the performance of model parallel and data parallel strategies under a 4-GPU setup to present a more thorough evaluation.
7. **Appendix C**: Extended the data augmentation experiments by increasing training epochs from 300 to 1000 to observe the fully converged performance and provide a detailed comparison and analysis.
8. **Appendix I**: Included new experiments on deactivated neurons, demonstrating that our mean goodness design resolves the issues caused by square goodness. Additionally, we analyzed the relationship between network depth and the deactivated neuron ratio, offering a deeper understanding and further validation of our approach.

We believe these revisions address the reviewers’ comments effectively and enhance the overall quality of the paper. Thank you again for your valuable feedback.

---

### Public Comment · ~Liang_Sun4 · 2025-04-21
**Descriptive modifications in the final version**

We made some minor descriptive revisions in the final version, which do not affect the conclusions of our method:

1. We updated the description of normalization in the Forward-Forward (FF) algorithm to "normalization by vector length" instead of "L2 norm," in order to ensure the accuracy of description. Although the original FF paper describes normalization as dividing a vector by its length (i.e., the L2 norm), and some public implementations (https://github.com/pytorch/examples/tree/main/mnist_forward_forward) and follow-up works (such as Symba https://arxiv.org/abs/2303.08418v1) adopt this approach, a Public Comment pointed out that the official code released by Hinton uses RMSNorm, a variant of LayerNorm. Since both L2 norm and RMSNorm aim to normalize vectors to a fixed magnitude, the term "normalization by vector length" better captures this shared goal and is thus more precise.

2. We also conduct a experiment in our ablation study where squared goodness is combined with RMSNorm. This showed improved performance compared to using L2 norm, but still lags significantly behind our proposed method. These results further support the effectiveness of our approach.

---

### Meta-Review · Area_Chair_zisx · 2024-12-20

**Metareview:**

This paper introduces a novel biologically plausible approach, which extends Forward-Forward training to deeper networks using layer normalization and mean goodness. The proposed method addresses limitations of Forward-Forward training by enabling more efficient training of non-trivial, deeper architectures.

After the rebuttal, this paper received three positive scores (8, 6, 6) and one negative score (5), with the primary concern focused on DeeperForward's performance. Overall, the paper provides a notable contribution to the field of Forward-Forward (FF)-based methods, and the AC recommends its acceptance.

**Additional Comments On Reviewer Discussion:**

Reviewer YDPM initially raised concerns regarding the training details, including the computational cost, memory-saving strategy, and parallel strategy of the proposed method, which were addressed after the rebuttal. The primary concern remains the performance of the FF-based method. However, the ACs acknowledge that while FF-based methods currently face limitations in achieving significant accuracy improvements, this does not diminish the research value of exploring this direction. Therefore, the recommendation is to accept the paper.

---

### Decision · Program_Chairs · 2025-01-22

Accept (Poster)